# First Isolation and Genomic Characterization of *Escherichia ruysiae* in Togo from a Five-Year-Old Patient with Gastroenteritis and Bloody Diarrhea

**DOI:** 10.3390/microorganisms13122694

**Published:** 2025-11-26

**Authors:** Kossi Kabo, Niokhor Dione, Kodjovi D. Mlaga, Tchadjobo Tchacondo

**Affiliations:** 1Laboratoire des Sciences Biomédicales, Alimentaires et de la Santé Environnementale, École Supérieure des Techniques Biologiques et Alimentaires, Université de Lomé, Lomé 01 BP 1515, Togo; fofokabo@gmail.com; 2Department of Bioengineering, University of Berkley, Berkeley, CA 94720, USA; ndione@berkeley.edu; 3The Microbiome and Mucosal Defence Research Unit, Institut de Recherche Clinique de Montréal, Montreal, QC H2W 1R7, Canada

**Keywords:** *Escherichia ruysiae*, WGS, *bla_EC-15_*, *senB*, *aatABC*, *eptA*, pediatric gastroenteritis, Clade IV *Escherichia*, Togo

## Abstract

*Escherichia ruysiae* is a recently characterized species within the *Escherichia* genus, often misclassified as *E. coli* due to limitations in existing operating procedures and diagnostic databases. In this study, we report the first isolation and genomic characterization of *E. ruysiae* in Togo, from a five-year-old female patient who was hospitalized with gastroenteritis and bloody diarrhea and subsequently died after eight days. Biochemical tests and MALDI-TOF initially identified the microorganism as *E. coli*, but phylogenomic and Average Nucleotide Identity (ANI) analysis confirmed it to be *E. ruysiae*, Clade IV with enteroaggregative associated genes. Whole genome sequencing of the strain FK53-34 enables the identification of resistance genes, including *bla_EC-15_*, *eptA*, and *pmrF*. The virulence profile of the strain included, but was not limited to *aap*, *aatABC*, and *senB* genes, which may support its pathogenicity and virulence. Multilocus sequence typing (MLST) did not match any known sequence type, which is obvious for a newly characterized microorganism. This study highlights the critical need for enhanced diagnostic tools and surveillance systems to identify emerging pathogens, including *Escherichia ruysiae*.

## 1. Introduction

The *Escherichia* genus is one of the most extensively studied bacterial groups in microbiology, largely due to its most prominent species, *Escherichia coli* (*E. coli*). It has served as a model organism for bacterial physiology, genetics, and host–microbe interactions for decades. Importantly, this genus encompasses both commensal and pathogenic members that colonize the gastrointestinal tract of humans and animals. Within the *E. coli* species itself, a remarkable diversity exists, ranging from harmless gut symbionts to strains responsible for severe gastrointestinal and extraintestinal infections, such as enterotoxigenic *E. coli* (ETEC), enterohemorrhagic *E. coli* (EHEC), enteroinvasive *E. coli* (EIEC), enteropathogenic *E. coli* (EPEC), enteroaggregative *E. coli* (EAEC), and diffusely adherent *E. coli* (DAEC). In addition, uropathogenic *E. coli* (UPEC) are a major cause of urinary tract infections, and meningitis-associated *E. coli* (MNEC) cause sepsis and meningitis. This duality makes the *Escherichia* genus especially compelling from both medical, veterinary, and ecological standpoints [1,2,3,4].

Six species have been described so far, including *E. coli*, *Escherichia hermannii*, *Escherichia fergusonii*, *Escherichia albertii*, *Escherichia marmotae*, *Escherichia whittamii*, and recently *E. ruysiae* [3,4,5,6,7,8,9,10]. All known *Escherichia* species are ubiquitous and potentially associated with animal [6] and human disease [5,6,7,8,9,10]. *E. ruysiae* was first isolated from the fecal sample of an international traveler, and all so far known to belong to two phylogroups (Cryptic Clade III and Clade IV) [9]. Several studies have shown that the phenotypic identification of *E. ruysiae* is confused with *E. coli*. This is because there is no standardized protocol in clinical microbiology to phenotypically differentiate *E. ruysiae* from the other members of the genus. Like other species of the genus *Escherichia*, *E. ruysiae* shows a genomic and proteomic signature very close to that of *E. coli*, but with certain differences that can be subtle and difficult to distinguish without a well-maintained database (for example, the Bruker Biotyper database, VITEK MS, or others) [6]. *E ruysiae* was spotted in a prevalence study of *Escherichia* spp., with zoonotic significance in urban crows in Japan, posing a significant risk for their potential transmission to humans [11]. The two cryptic clades described so far, Clades III and IV, are commonly found in animals and the environment. Studies have investigated the ubiquity of *E. ruysiae* and commensalism in the human intestinal tract [6]. Recently, *E. ruysiae* was isolated in 2024 from wild boars in Italy; therefore, its prevalence in livestock needs further investigation [12]. Also, antimicrobial resistance in *E. ruysiae* has been shown to be associated with conjugative plasmids carrying clinically relevant antibiotic resistance genes, and it is thought to serve as a reservoir of resistance genes globally [13]. To date, the pathogenicity of *E. ruysiae* is not known and has not been demonstrated in any clinical or veterinary cases. Also, this bacterium is generally overlooked in clinical and veterinary diagnostics globally because it lacks a standardized differential phenotypic assay in microbiology to delineate it from *E. coli.* In addition, its spectrum is still not in any of the MALDI ToF databases [14]. This makes the majority of *E. ruysiae* identified as *E. coli* with a comfortable score.

In Togo, the microbiological diagnosis is usually limited to phenotypic identification assays after the growth and isolation of the microorganism in pure colonies. Using these conventional techniques, a case of gastroenteritis was reported to be caused by *E. coli* from a bloody stool sample collected from a five-year-old child hospitalized in a district medical center in the maritime region of Togo. For the first time, we performed a thorough genomic description of the strain and demonstrated that this was *E. ruysiae* instead of *E. coli*. In this study, we aim to describe the phenotypic and genomic characteristics of this strain and highlight specific genomic characteristics associated with the virulence and antimicrobial resistance.

## 2. Materials and Methods

### 2.1. Sample Collection

Two swabs were taken from the stool sample collected in clean containers, equipped with a tight-fitting, airtight lid. Each swab was introduced into Cary-Blair medium (OXOID, Basingstoke, UK), previously cooled with ice for one to two hours before stool sample collection. The swabs, loaded with stool (by rotating the swab into the stool or anus), were introduced into the transport medium for transportation. The upper part of the stem, touched by fingers, was cut off and discarded. The caps of the Cary-Blair tubes and the lids of the stool containers were screwed back on, and the tubes were labeled following the protocol described by March and Ratnam [15]. The samples were transported to the laboratory in a refrigerated cooler box to be processed. The remaining waste was destroyed following the guidelines by standards of biohazard type 2 [16,17].

### 2.2. Isolation and Identification

The physical and microscopic examination of stools was carried out following the standard operational procedure (SOP) of the clinical microbiology laboratory [18,19]. A loopful of the stool sample was inoculated into peptone enrichment broth at a 1:10 sample-to-broth ratio and incubated overnight at 37 °C. Subsequently, a 10 µL loopfull of the overnight enrichment broth was plated onto SMAC medium (Sorbitol-MacConkey) and incubated at 37 °C for 24 h [19]. Pink colonies were further plated on Eosin Methylene Blue (EMB) agar and incubated at 37 °C for 24 h [20]. Colonies were counted based on shape, and the dominant colonies with metallic sheen, suggestive of *E. coli*, were confirmed with phenotypic assay using Api20E read by commercially available Microbact GNB 24E (OXOID, Basingstoke, UK) according to the manufacturer’s instructions. The confirmation Api20E assay was carried out with the strain FK53-34 at Institut Pasteur (Paris, France) using a Biomerieux kit (bioMérieux SA, Marcy-l’Etoile (Lyon), France). A 100 µL of 0.5 McFarlan bacterial suspension was inoculated into each well, and the strips were then incubated. After 2 h of incubation at 37 °C, a change in the media (or after adding revelers) was used to identify the profile of the bacterium [15]. In addition, we used MALDI ToF MS (matrix-assisted laser desorption ionization time of flight mass spectrometry) to confirm or identify pure growth isolates [21,22].

### 2.3. Susceptibility Testing

From a suspension of 0.5 McFarland, prepared from 2 to 3 pure colonies of FK53-34 on Mueller Hinton medium [23], we carried out a susceptibility test according to Comite d’antibiogramme de la Societe francaise de microbiologie guideline (CA-SFM). Disks of 5 µg of Ciprofloxacin (CIP), 30 µg of Cefepim (FEP), 30 µg of Aztreonam (ATM), 5 µg of Cefotaxim (CTX), 75 µg of Ticarcillin (TIC), 75 µg/10 µg of Ticarcillin + clavulanic acid (TCC), 30 µg of Cefoxitin (FOX), 30 µg of Cefuroxim (CXM), 10 µg of Ceftazidim (CAZ) [24], 75 µg of Ticarcillin (TIC), 20 µg of Amoxicillin (AMX), 30 µg of Nalidixic Acid (NAL) [25] were tested. Periodical AST control was performed following standard protocols, including the use of a recognized quality control (QC) strain (*E. coli* ATCC 25922) was used as quality control to ensure the validity of the results. Inhibition zones obtained for the QC strain are validated before proceeding with any AST interpretation.

The inhibition diameters are interpreted as resistant, intermediate, and susceptible according to reference values (diameters in mm) for disk susceptibility testing (agar disk diffusion) recommendations [24,25].

### 2.4. DNA Extraction and Sequencing

We extracted nucleic acid of FK53-34 at Institut Pasteur from 24 h culture growth on Petri dishes using the KingFisher technique (Themo Scientific, OXQ32853, Oxoid brand, Wesel, Germany), according to the manufacturer’s instructions. After quality control and DNA quantification using Nanodrop spectrophotometry and a Qubit fluorometer [26], the DNA was denatured. By adding 500 μL of prepared 1.4 pM libraries into the reservoir, we diluted and prepared DNA fragment ends for adapter ligation and attached Illumina sequencing adapters to the DNA fragments. We amplified adapter-ligated fragments to enrich the library [27]. Sequencing libraries were constructed, and the libraries were quantified using the 2100 Bioanalyzer System (Agilent Technologies, Inc., Santa Clara, CA, USA) and Kapa Sybr Fast qPCR Kit (Kapa Biosystems (part of Roche, Cape Town (Old Warehouse Building, Black River Park, Fir Road, Observatory, South Africa))). We carry on the sequencing on an Illumina Miseq V3 platform using a 2 × 300 paired-end approach (Illumina Inc., San Diego, CA, USA) [28].

### 2.5. Assembly and Annotation

We assessed the quality of the raw reads of FK53-34 using FastQC v0.11.5 [29] and filtered them using Trimmomatic v3 with a cut-off of Q28 [28]. Redundant or over-represented sequencing reads were eliminated, sequencing errors were corrected, and only reads meeting stringent nucleobase quality criteria (Phred score > 28) were retained for downstream analyses. The high-quality reads of FK53-34 were assembled with a de novo assembly approach using Spades V3.13.1 [30], and the final assembly scaffolds were checked using Checkm2 v1.0.2 [31] to provide accurate estimates of genome completeness, contamination, and heterogeneity. The taxonomy classification was carried out using GTDB-Tk v2.4.0 (RefDB version r220) [32]. A contig rearrangement was carried out using Mauve progressive alignment and *E. ruysiae* (strain OPT1704, AB136). We performed the annotation of the obtained scaffolds using Prokka1.14.5 [31]. We used Get Homologues [33] to compute the coding sequence (CDS) ANI of FK53-34 using the OrthoMCL version 2.0.9 approach with close *E. ruysiae* genomes available on NCBI. Furthermore, we performed a core alignment of the all available *E. ruysiae*, including FK53-34, using Scapper (https://github.com/tseemann/scapper.git), accessed on 29 March 2025, a reference-based alignment tool composed of MuMmer 4 [34], and TrimAL [35]. To support the ANI cladogram, we constructed a core genome maximum-likelihood phylogenetic tree using raxml-ng [36] and used FigTree [37] to visualize the tree. Proksee [38] was used to visualize the assembly, the annotation, the virulome, and the resistome of our genome.

### 2.6. Resistome, Virulome Genome Sequence Typing

We characterized the resistome and the virulome of FK53-34 using Abricate v1.9.8 (GitHub. https://github.com/tseemann/abricate, accessed on 23 June 2025) [39]. The resistance genes were searched against the Resfinder 4.1 database (367 resistance genes) [40], NCBI (6570 sequences downloaded on 23 June 2023), CARD V3.2.4 (6627 ontology terms, 5010 reference sequences, 1933 mutations, 3004 publications, and 5057 AMR detection models [41]), ARG-ANNOT V6 (1749 sequences) [42], and MEGAres V3.0 (9000 resistance genes antimicrobials) [43]. We identified virulence genes by mapping the genome sequences against the virulence database *VFDB* V1.0 using default parameters [44]. We performed in silico typing using Ectyper V0.8.1 [45] with default parameters, and sequence types (ST) were determined using MLST 2.23.0 [46,47] based on seven housekeeping genes (*adk*, *fumC*, *gryB*, *icd*, *mdh*, *purA*, and *recA*). We finally used the phylogroup classification tool, ClermonTyping [48], to identify the phylogroup of our strain.

## 3. Results

### 3.1. Clinical Case Description

A 5-year-old, 18 kg female was examined by a general practitioner for abdominal pain with vomiting and bloody diarrhea without fever. The diagnosis was “gastroenteritis”. The present clinical case was found at Centre Medico-Social de Togblekope (altitude: 71 m above sea level, coordinates: 6°16′24″ N and 1°12′49″ E), located in the suburban region of Lomé. Clinical symptoms are dominated by vomiting, with bloody diarrhea and absence of fever and abdominal cramps. The patient was first treated with Motilium^®^ (domperidone) oral suspension (2/day), ceftriaxone (500 mg) injectable (2/day), and Actapulgite^®^ (activated Mormoiron attapulgite) powder in oral suspension (2 days). The vomiting stopped on the third day; however, the bloody diarrhea and abdominal cramps persisted. On the fourth day, symptoms reappear with a thermal surge up to 40 °C in the afternoon. Most of the blood cell counts are within normal range (red count = 4.22 × 10^6^/µL (4.8–6.2 × 10^6^), white count = 6200 cells/µL, hemoglobin = 13.6 g/dL (11–15 g/dL), hematocrit = 43.8 (40–54), platelet = 244,000/µL (150,000–400,000/µL)). On the seventh day, the results of the microbiology test and mass spectrometry (MALDI TOF) of FK53-34 showed *E. coli* with a score of 2.2 and sensitivity to all antibiotics tested. The patient was later referred to a gastroenterologist around 10 a.m., but she could not survive. She died after 8 days of hospitalization, despite the care provided. The autopsy concluded at 2 p.m. that immense dehydration from infectious gastroenteritis and weight loss were estimated at nearly 14% of total weight.

### 3.2. Phenotypic, Biochemical Characterization and Resistance Profile

A culture growth of the strain FK53-34 at 37 °C in an aerobic atmosphere for 24 h on the TSA agar as transparent, refractive, and translucent colonies of various sizes. The colonies appear creamy to white, with circular shapes and edges ranging from smooth to slightly convex. They vary in size, from 1 to 3 mm in diameter. FK53-34 is Gram-negative (Appendix A). Biochemical tests performed on the API 20E assay showed results in Table 1. The strain FK53-34 shows a similar phenotypic metabolism profile to that of the reference AB134, with the exception that the strain FK53-34 metabolizes citrate (CIT), tryptophane (TDA), and melibiose (MEL). Compared to *E. coli*, the latest was positive for alanine dehydrogenase (ADH), lysine dehydrogenase (LDC), and saccharose (SAC), while *E. ruysiae* was negative.

FK53-34 is phenotypically susceptible to all antibiotics tested according to reference values (diameters in mm) for disk susceptibility testing (disk diffusion method) recommendations [25].

### 3.3. Genomics Features and Challenges in the Strain Identification

We assembled the genomes of FK53-34 into 4,785,617 bp, comprising 136 contigs with a GC content of 50.55% with an average length of the contigs ≈ 35,190 bp. The whole-genome sequence of strain FK53-34 has been deposited in GenBank under BioProject accession number PRJNA1223433. The N50 is 94,629 with a coding density of 0.87. There were 4464 coding sequences (CDS), 4558 predicted genes, 83 tRNA, 1 repeat region, 1 mmRNA, 10rRNA, 1173 markers, and no heterogeneity (Appendix A). The ANI (Average Nucleotide Identity) gives an average similarity of nucleotide sequences of 98.78 for *Escherichia ruysiae*. The MultiLocus Sequence Typing (MLST) match with the following allele: adk(356), gyr(73), icd(670), mdh(56), purA(322), recA(454), and no best match with fumC(~1709). The final ST ID as well as the antigenic profile, which could not identify alleles for the genotype ST and OH antigens, respectively. However, the serogroup of the strains FK53-34 was identified as Clade IV. The strain FK53-34 is close to the strain C14-1 and S1-IND-07-A with ANI 99.04% and 98.91%, respectively (Figure 1). We found two clusters in the ANI dendrogram, which was confirmed by the core genome maximum likelihood phylogenetic tree grouped in Clade III and Clade IV (Figure 2)

### 3.4. Resistome, Virulome, and Mobilome of FK53-34

*E. ruysiae* have been found to harbor antimicrobial resistance genes, including those encoding extended-spectrum beta-lactamases (ESBLs). FK53-34 carries genes associated with the production of beta-lactamases *BlaEC*, *(Bla)*_CTXM-15_, *(Bla)ampH*, *(Bla)AmpC1*, *(Bla)AmpC2*, and an envelope stress-responsive two-component system (CpxAR). It carries the *eptA* and *pmrF* genes, conferring resistance to cationic antimicrobial peptides, such as colistin, and multidrug resistance genes (*emr* and *edt*). The most important resistance genes found are shown in Table 2. The raw blast search is provided as Appendix A. Several virulence factor genes were also identified, including those encoding flagellar protein (*flgABCDE*, *filABC*, *flhABC*, and *motAB*), colonization factors (*fimA*, *fimB*, *fimC*, *fimD*, *fimE*, *fimF*, *fimH*, and *fimI*), fitness (*iutA* and *kpsMII*), heat-stable enterotoxin (*astA*, *senB*), chemotaxis regulatory protein (*CheR*, *CheB*, and *CheY et CheZ*), AidA-I adhesin-like protein (*ehaAB*, *hofB*, *hofC*, and *hofq*), *ompT* gene for outer membrane protease, and plasmid R100 complement resistance gene (*traJ* and *traT*). We also note the presence of the operon (*aatPABC*), adaptation gene *(cfa*), adhesion and biofilm formation gene (*csgABCDEFG* and *agn43*), and blood–brain barrier invasion (*ibeBC*) (Table virulence factor). The virulome shows the absence of the wzx and wzy genes. We noted also the presence of specific adhesion genes (*EC958_4611*, *EC958_4614*), immune evasion (ECS88_3547), motility and biofilm (*fliC-H56*), and pathogenicity islands (*EC55989_3335* and *Z1307*) in the FK53-34 isolate. Virulence genes are commonly associated.

Pathogenicity in *E. coli* is summarized in Table 3. The raw blast search is provided in Appendix A.

## 4. Discussion

The successful isolation and culturing of *Escherichia ruysiae* from the fecal sample of a five-year-old patient with gastroenteritis and bloody stools in Togo significantly expands our understanding of its ecology and its pathogenic potential. Previously, *E. ruysiae* has been detected in diverse hosts and locations, including humans in Europe [6], an urban wild crow in Asia [11], a domestic chicken in North America [50], and domestic dogs in Oakland, USA [51], highlighting its broad ecological niches. The presence of *E. ruysiae* in humans, animals, and the environment confirms its contribution to the One Health concept that recognizes the interconnectedness of human, animal, and environmental health. These findings underscore the need for phylogenomic studies of *E. ruysiae* to clarify its evolutionary relationships and global distribution as an emerging pathogen. Many studies have noted the challenge of correctly identifying *E. ruysiae*, often misclassifying it as *E. coli*, particularly when using MALDI-TOF MS version 4.1.70 [51,52]. This misidentification can be explained by the absence of *E. ruysiae* specific spectra in current databases, underscoring the urgent need to update reference libraries. In addition, we noticed that the biochemical profile of *E. ruysiae* is neither consistent nor stable. There were some differences between *E. ruysiae* FK53-34 and the reference strains (*E. ruysiae* AB136). *E. ruysiae* FK53-34 can ferment and metabolize citrate (CIT), tyrosine arylamidase (TDA), and melibiose (MEL), while others were unable. This showed that traditional biochemical assays lack sufficient discriminatory power, especially among closely related bacterial taxa [53]. Improved diagnostic accuracy generally requires confirmation from molecular or genomic methods, and indeed, several isolates initially classified as *E. coli* were later correctly identified as *E. ruysiae* following whole-genome sequencing [51]. Hence, these biochemical differences, while potentially useful, should be interpreted cautiously until further validated by molecular biology techniques or genomics. The ANI heatmap confirmed that the strain FK53-34 is definitively not *E. coli*; this is supported by the core genome phylogenetics analysis of *E. ruysiae* isolated from various environments, which reveals notable genetic diversity divided into two branches associated with the Clade III [6] and the clade IV [13]. The FK53-34 strain belongs to the clade IV clusters, and some of the strains in that cluster have been described to harbor antimicrobial resistance genes and are found in diverse environments. Our data shows that all analyzed *E. ruysiae* carry the gene *blaEC-15*, which supports the previous finding that *E. ruysiae* might serve as a reservoir of antimicrobial resistance genes [13]. For instance, FK53-34 shares a close phylogenetic relationship with C14-1, a strain previously isolated from a healthy domestic hen in the UK [50], reinforcing the cross-host adaptability of *E. ruysiae*. Unlike *E. coli*, which can typically be classified into phylogroups (A, B1, B2, D, etc.) [50], *E. ruysiae* does not fit into this classification. This is supporting the fact that our isolate could not be assigned to a known sequence type (ST).

Genomics plays a crucial role in modern clinical microbiology, particularly in taxonomic classification, profiling antimicrobial resistance (AMR) carriage, and deciphering their mechanisms. Despite phenotypic susceptibility to all antibiotics tested, genomic analysis shows the presence of multiple resistance associated genes (including efflux pumps associated with resistance mechanisms), such as *blampH* and many other chromosome-mediated resistance genes (*blaEC-15*, *eptA*, *pmrF*), consistent with silent/low-expression resistomes and the known requirement for regulatory activation of polymyxin-resistance pathways (*PmrAB/PhoPQ*) for lipid A modification (*eptA/pmrC*, *arn/pmrF*) to confer resistance. In the absence of activating mutations or promoter insertions, these loci may not elevate MICs [54,55]. Although the *blaEC-15* gene was detected in strain FK53-34 and promoter motifs seemed to be intact, we have not confirmed this discrepancy by RT-PCR to confirm whether the gene can be expressed or not, which could potentially explain the absence of phenotypic β-lactam resistance. Future work will aim to include the phenotypic testing of colistin and/or polymyxin B to assess the potential activity of the *eptA* and *pmrF* genes. The gene *blaEC-15* is a natural non-plasmid transferable chromosomal variant of *AmpC-like*. Expression of chromosomal *blaEC* (*AmpC-like*) alleles can be insufficient for cephalosporin resistance without a functional promoter [54,55]. This might explain why the strain was sensitive to all cephalosporins tested, including Cefepim, Cefotaxim, Cefoxitin, Cefuroxim, and Ceftazidim. This has been previously shown with *E. coli*, which harbored genes associated with resistance to antimicrobial but with a susceptible phenotypic profile when tested [49,56,57,58]. Although we did not perform PCR or Sanger sequencing to validate antimicrobials resistance and virulence genes, the whole-genome sequencing assembly meets and exceeds required quality thresholds. Previous studies have shown that a genome coverage of ≥15–20× is sufficient to accurately detect ARGs in assembly-based workflows [59]. Furthermore, in silico detection utilizing identity thresholds of 90% and adequate sequence coverage has been validated against PCR benchmarks in similar contexts [60]. In clinical pathogen genomics, a coverage of ≥30× is generally accepted as ensuring high confidence in marker detection and minimizing false assignments [1,2,61]. Given that our data met these standards, we consider our results to be both robust and reproducible.

Of all *E. ruysiae* published so far, only the strain FK53-34 carries specific virulence genes, namely *aap/aspU*, *aatABCP*, *senB*, and *traJT* generally found in *E. coli*. The gene *aap/aspU*, also known as Dispersin, is thought to neutralize the bacterial cell surface, repelling and projecting the positively charged AAFs (aggregative adherence Fimbriae), thereby facilitating the dispersal of the bacteria after initial aggregation and spread across the intestinal mucosa. This process also supports biofilm formation and chronic colonization, contributing to the persistent diarrhea typical of EAEC infections. This dispersal may allow the bacteria to spread and establish new foci of infection [62]. The *AatABCD* export system is a critical exporter responsible for secreting specific proteins from the bacterial cytoplasm to the extracellular environment. Its most well-known role in EAEC is the export of the dispersin (*Aap/AspU*) protein [63]. The *senB* heat stable enterotoxin gene encodes an enterotoxin that increases fluid and electrolyte secretion in the intestinal mucosa. It contributes to watery diarrhea, complementing other virulence factors such as the type III secretion system and invasions [64]. All these genes, if expressed, may have exacerbated the diarrhea condition of the patient, extensively lost water, and inevitably led to death. In addition, the presence of *ibeB* and *ibeC* genes is known to confer to the pathogen the ability to cross the blood–brain barrier, producing the invasion of brain endothelial cells, and promoting neonatal meningitis [65,66]. Although several virulence-associated genes (e.g., *aap/aspU*, *aatABCP*, *senB*, and *traJT*) were identified in FK53-34, their direct contribution to the clinical outcome cannot be established from this study. The absence of functional validation (e.g., histopathology, in vivo models, or gene expression studies) represents a limitation. Future research should therefore assess the pathogenic role of these factors through experimental infection models and transcriptomic analyses [67].

Although the gastroenteritis looked mild and should have been treated easily with common antibiotic therapy, the misidentification of the strains, the enclosed antimicrobial resistance profile added to the arsenal of virulence factors have contributed to the overlooking of the diagnosis, which led unfortunately to the death of the patient.

## 5. Conclusions

This study reports the first clinical isolation and genomic characterization of *Escherichia ruysiae* in Togo, associated with a fatal outcome of gastroenteritis in a pediatric patient. Our findings reveal that *E. ruysiae* harbor antimicrobial resistance genes not detected in the susceptibility test and a virulome mainly found in pathogenic *E. coli*. These results underline the diagnostic limitations of conventional methods in accurately identifying cryptic *Escherichia* species and highlight the potential public health threat posed by overlooked pathogenic strains. The genomic features of FK53-34 reinforce the need for expanding surveillance, updated diagnostic tools, and incorporation of *E. ruysiae* into routine microbial monitoring. From a One Health perspective, its presence in both clinical, veterinary, and environmental settings points to the urgent need to integrate approaches and control antimicrobial resistance of emerging pathogens.

## Figures and Tables

**Figure 1 microorganisms-13-02694-f001:**
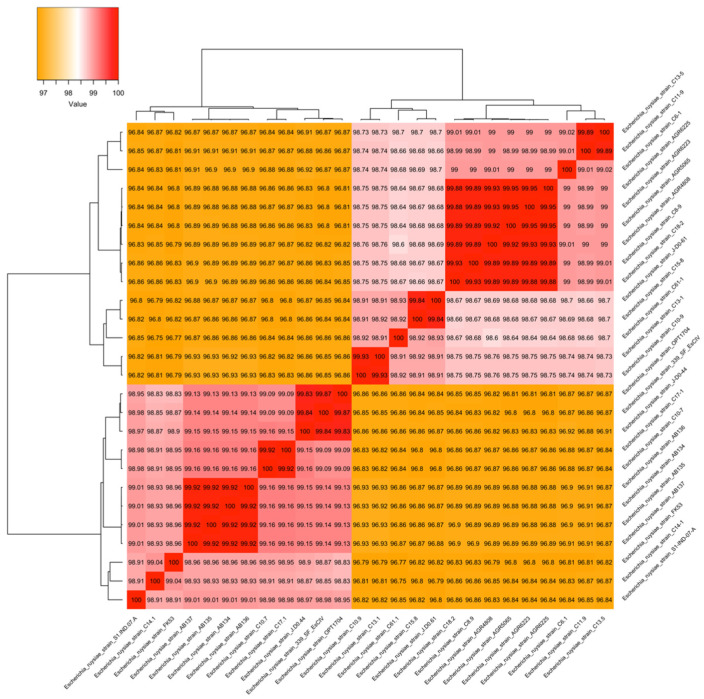
**Heatmap of ortholog CDS Average Nucleotide Identity (ANI) of *E. ruysiae* FK53-34 compared to other publicly available *E. ruysiae* as of 24 February 2025.** The ANI was computed using the GET HOMOLOGUE version 3.7.4 package with default parameters [35]. The heatmap shows two clusters, as demonstrated by Ellington, 2017 [49] and the strain FK53-34 belongs to the clade IV cluster and is close to the strains C14.1 and S1.IND.07.A confirming the new classification of the FK53-34 as *E. ruysiae*.

**Figure 2 microorganisms-13-02694-f002:**
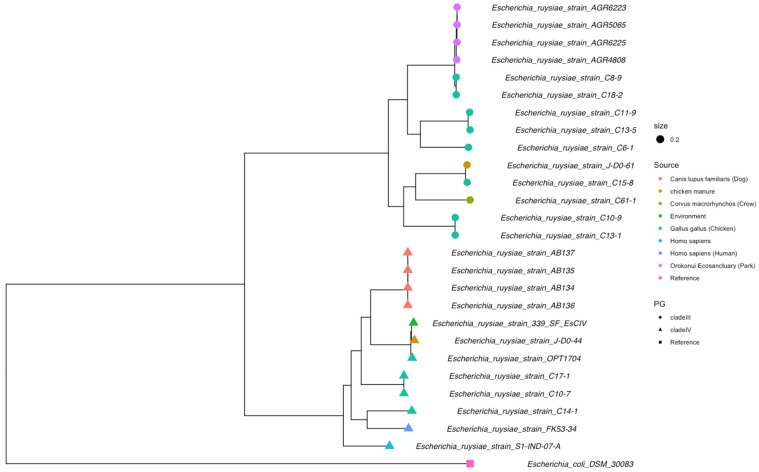
**Maximum likelihood phylogenetic tree of the core genome of all available *E. ruysiae*, including the strain FK53-34**. The tree is mid-rooted in decreasing topology and shows two clades, III (round-shaped tips) and IV (triangle-shaped tip) and colored by source of isolation, the strain FK53-34 belongs to the clade IV clusters.

**Table 1 microorganisms-13-02694-t001:** Biochemical reaction assay of the FK53-34, *E. ruysiae AB134*, and *E. coli* strain.

Tested Substrates	FK53-34 (This Study)	*E. ruysiae* [6]	*E. coli* [11]
ONPG	+	+	+
ADH	−	−	+
LDC	−	−	+
ODC	+	+	+
CIT	+	−	−
H2S	−	−	−
URE	−	−	−
TDA	+	−	−
IND	−	+/−	+
VP	−	+	−
GEL	−	−	−
GLU	+	+	+
MAN	+	+	+
INO	−	−	−
SOR	+	+	+
RHA	+	+	+
SAC	−	−	+
MEL	+	−	−
AMY	−	−	−
ARA	+	+	+

ONPG—Ortho-Nitrophenyl-β-galactoside test (tests for β-galactosidase activity), ADH—Arginine Dihydrolase (tests ability to hydrolyze arginine), LDC—Lysine Decarboxylase (tests ability to decarboxylate lysine), ODC—Ornithine Decarboxylase (tests ability to decarboxylate ornithine), CIT—Citrate utilization test (ability to use citrate as sole carbon source), H2S—Hydrogen Sulfide production, URE—Urease (tests ability to hydrolyze urea), TDA—Tryptophan Deaminase (aDeaminase), IND—Indole production (tests ability to produce indole from tryptophan), VP—Voges-Proskauer test (detects acetoin production), GEL—Gelatinase (tests hydrolysis of gelatin), GLU—Glucose fermentation, MAN—Mannitol fermentation, INO—Inositol fermentation, SOR—Sorbitol fermentation, RHA—Rhamnose fermentation, SAC—Sucrose fermentation, MEL—Melibiose fermentation, AMY—Amygdalin fermentation, ARA—Arabinose fermentation.

**Table 2 microorganisms-13-02694-t002:** Essential antimicrobial genes found in *E. ruysiae*.

Isolate	*aadA1-pm*	*bla_LAP-2_*	* _qnrS1_ *	*sul2*	*tet*(A)	*bla* _CTX-M-15_	*bla_EC-15_*	*bla_EC-8_*
ABC134	−	−	−	−	−	−	+	−
ABC135	−	−	−	−	−	−	+	−
ABC136	−	−	−	−	−	−	+	−
ABC137	−	−	−	−	−	−	+	−
C10-7	−	−	−	−	−	−	+	−
C10-9	−	−	−	−	−	−	+	−
C11-9	−	−	−	−	−	+	+	−
C13-1	−	−	−	−	−	−	+	−
C13-5	−	−	−	−	−	−	+	−
C14-1	−	−	−	−	−	−	+	−
C15-8	−	−	−	−	−	+	+	−
C17-1	−	−	−	−	−	−	+	−
C18-2	−	−	−	−	−	−	+	−
C6-1	−	−	−	−	−	+	−	+
C61-1	−	−	−	−	−	−	+	−
C8-9	−	−	−	−	−	−	+	−
OPT1704	−	+	+	+	−	+	+	−
S1-IND-07-A	+	−	+	+	+	+	+	−
FK53-34	−	−	−	−	−	−	+	−

**Table 3 microorganisms-13-02694-t003:** Distribution of virulence genes found only in strain FK53-34.

Isolates	*aap/aspU*	*aatA*	*aatB*	*aatC*	*aatP*	*agn43*	*senB*	*traJ*	*traT*
ABC134	−	−	−	−	−	−	−	−	−
ABC135	−	−	−	−	−	−	−	−	−
ABC136	−	−	−	−	−	−	−	−	−
ABC137	−	−	−	−	−	−	−	−	−
C10-7	−	+	−	−	−	−	−	−	−
C10-9	−	+	−	−	−	−	−	−	−
C11-9	−	+	−	−	−	−	−	−	−
C13-1	−	+	−	−	−	−	−	−	−
C13-5	−	+	−	−	−	−	−	−	−
C14-1	−	−	−	−	−	−	−	−	−
C15-8	−	−	−	−	−	−	−	−	−
C17-1	−	+	−	−	−	−	−	−	−
C18-2	−	−	−	−	−	−	−	−	−
C6-1	−	+	−	−	−	−	−	−	−
C61-1	−	−	−	−	−	−	−	−	−
C8-9	−	−	−	−	−	−	−	−	−
OPT1704	−	−	−	−	−	+	−	−	−
S1-IND-07-A	−	−	−	−	−	−	−	−	−
FK53-34	+	+	+	+	+	+	+	+	+

*aap*: Anti-aggregation/Dispersin, *aatA*, *aatB*, *aatC*, *aatP*: Autotransporter Adhesin, *agn43*: Antigen 43 (secretive system), *senB*: Secreted Enterotoxin (heat stable enterotoxin), *traJ*, *traT*: transporter, pili.

## Data Availability

The genome of the FK53-34 has been submitted to NCBI, Bioproject: PRJNA1223433 with accession number JBLRVR000000000.

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
