# Peer review of "First Isolation and Genomic Characterization of Escherichia ruysiae in Togo from a Five-Year-Old Patient with Gastroenteritis and Bloody Diarrhea"

_microorganisms, 2025, doi:10.3390/microorganisms13122694_

Round 1

Reviewer 1 Report (Previous Reviewer 1)

Comments and Suggestions for Authors

I note that this is a resubmitted manuscript. The authors have made the necessary revisions based on the previous review comments. The quality of the manuscript has been significantly improved in its current form, and I find it acceptable for publication in Microorganisms (ISSN 2076-2607). I have no further comments or suggestions.

Author Response

Reviewer 1

I note that this is a resubmitted manuscript. The authors have made the necessary revisions based on the previous review comments. The quality of the manuscript has been significantly improved in its current form, and I find it acceptable for publication in Microorganisms (ISSN 2076-2607). I have no further comments or suggestions.

We thank the reviewer for the positive assessment of our revised manuscript and his recommendation for publication in Microorganisms. We greatly appreciate the time and effort dedicated to reviewing our work and are pleased that the revisions met the reviewer’s expectations.

Reviewer 2 Report (Previous Reviewer 2)

Comments and Suggestions for Authors

This study presents only a basic genomic analysis of a single Escherichia ruysiae isolate, including genome sequencing and analysis of antimicrobial resistance, along with genomic screening for resistance and virulence genes.

  1. Figure 1 is not visible in the main text.
  2. Figure 2 would benefit from additional metadata for the included strains, such as geographic origin and host source, to improve contextual interpretation.
  3. The manuscript contains numerous minor errors and inconsistencies. For example, on lines 22–23, the gene pmrF is listed simultaneously as both a resistance gene and a virulence gene.

Author Response

Reviewer 2

This study presents only a basic genomic analysis of a single Escherichia ruysiae isolate, including genome sequencing and analysis of antimicrobial resistance, along with genomic screening for resistance and virulence genes.

  1. Figure 1 is not visible in the main text.

New figure is generated and will submitted separately

  1. Figure 2 would benefit from additional metadata for the included strains, such as geographic origin and host source, to improve contextual interpretation.

New figure is generated and will submitted separately, we improved comments accordingly

  1. The manuscript contains numerous minor errors and inconsistencies. For example, on lines 22–23, the gene pmrF is listed simultaneously as both a resistance gene and a virulence gene.

 We thank the reviewer. This is a type error and it is corrected

Reviewer 3 Report (Previous Reviewer 3)

Comments and Suggestions for Authors

Line 21, 27, and elsewhere: bla EC-15 with EC15 as subscript

Lines 21, 22:                                How can pmrF be a resistance gene and a virulence gene? Please check and correct.

Line 32:                                         Insert a space between E. and coli ((E. coli).

Line 40:                                         Please use “In addition,” rather than “Additionally” at the beginning of a sentence.

Line 56:                                         spp. not in italics.

Line 65:                                         Please replace hasn’t by has not

Lines 68, 106:                              MALDI ToF

Line 83:                                         (OXOID, UK) not in italics

Line 102:                                       Institut Pasteur (Paris) not in italics

Line 110,111                                Comite de l’antibiogramme de la societe francaise de microbiologie

Line 118:                                       ... as quality control …

Line 122:                                       One of the general rules – please see https://doi.org/10.1186/s44280-023-00024-w - is that a performance standard and the interpretive criteria mentioned therein, represent an entity. That means that you cannot perform antimicrobial susceptibility testing according to CA-SFM and evaluate the results according to CLSI or EUCAST.

Line 140:                                       de novo

Line 184:                                       could not

Line 231:                                       In the pdf file that I downloaded, there was no Fig. 1

Line 238:                                       Bacterial genus and species designations in Fig. 2 should be in italics.

Line 244:                                       Beta-lactamase designations are still not correct. The CpxAR is not a beta-lactamase, but an envelope stress-responsive two-component system.

Line 246:                                       (emr, mdt) with emr and mdt in italics.

Line 260:                                       E. ruysiae in italics

Table 2:                                         Delete: (AGly), -pm, (Bla), (Flq), (Sul), (Tet), LAP-2 as subscript, EC-15 as subscript, EC-8 as subscript, closing bracket  in tet(A) not in italics.

Author Response

Reviewer 3

Line 21, 27, and elsewhere: bla EC-15 with EC15 as subscript

It was corrected. Thanks

Lines 21, 22: How can pmrF be a resistance gene and a virulence gene? Please check and correct.

Refer to reviewer 1’s comment. It was corrected. Thanks

Line 32: Insert a space between E. and coli ((E. coli).

Thank you to the reviewer for his insightful remark. It is done

Line 40: Please use “In addition,” rather than “Additionally” at the beginning of a sentence.

We thank the reviewer. It is done

Line 56: spp. not in italics.

Done

Line 65: Please replace hasn’t by has not

We thank the reviewer for his comment. It is replaced

Lines 68, 106: MALDI ToF

Done

Line 83: (OXOID, UK) not in italics

Done

Line 102: Institut Pasteur (Paris) not in italics

Done

Line 110,111: Comité de l’antibiogramme de la société française de microbiologie

Done

Line 118:  ... as quality control …

Done

Line 122: One of the general rules – please see https://doi.org/10.1186/s44280-023-00024-w - is that a performance standard and the interpretive criteria mentioned therein, represent an entity. That means that you cannot perform antimicrobial susceptibility testing according to CA-SFM and evaluate the results according to CLSI or EUCAST.

Both references were used for the interpretation and are now cited in the manuscript

Line 140:  de novo

Done

Line 184: could not

Done

Line 231: In the pdf file that I downloaded, there was no Fig. 1

The figure is updated. Thanks

Line 238:  Bacterial genus and species designations in Fig. 2 should be in italics.

The bacterial genus and species designations are now formatted in italics

Line 244: Beta-lactamase designations are still not correct. The CpxAR is not a beta-lactamase, but an envelope stress-responsive two-component system.

It is formatted in the main manuscript. Thanks

Line 246: (emr, mdt) with emr and mdt in italics.

Done

Line 260: E. ruysiae in italics

Done

Table 2:  Delete: (AGly), -pm, (Bla), (Flq), (Sul), (Tet), LAP-2 as subscript, EC-15 as subscript, EC-8 as subscript, closing bracket  in tet(A) not in italics.

Done

This manuscript is a resubmission of an earlier submission. The following is a list of the peer review reports and author responses from that submission.

Round 1

Reviewer 1 Report

Comments and Suggestions for Authors

The authors report the isolation and genomic characterization of Escherichia ruysiae (FK53-34) in Togo, isolated from a five-year-old female patient hospitalized with gastroenteritis and bloody diarrhea who succumbed after 8 days. Whole-genome sequencing revealed antimicrobial resistance genes, diverse aminoglycoside-modifying enzymes, and virulence genes of this strain. This study underscores the urgent need for enhanced diagnostic capabilities and surveillance systems to detect emerging pathogens like E. ruysiae. While the study provides valuable insights, several issues regarding methodological clarity, data interpretation, and presentation require careful revision prior to publication consideration.

1. The pathogenic strain FK53-34 was isolated from a 5-year-old, 18 kg female. Did this girl have any underlying medical conditions or complications? Apart from FK53-34, were any other pathogenic microorganisms detected in the sample? Was the sample collection conducted with the consent of the child’s parents or guardians?

2. In the title of the manuscript, "First Genomic Characterization of Escherichia ruysiae...". The authors need to consider whether this statement is rigorous.

3. In the first paragraph of the Introduction, the full name Escherichia coli and its abbreviation E. coli have already been introduced. Please replace subsequent instances of Escherichia coli with the abbreviation E. coli (e.g., Lines 45–47, 100, 179).

4. Lines 61-62: Recently, E. ruysiae was isolated in 2024 from wild boars in Italy, therefore, its prevalence in livestock needs to be further investigated. I suggest this sentences be corrected to: E. ruysiae was isolated in 2024 from wild boars in Italy; therefore, its prevalence in livestock needs further investigation.

5. Some numbers and units lack a half-width space between them, e.g., 30µg (Lines 113, 115, 116), 75µg (Line 114), 10µg (Line 116). In scientific writing, a space should typically separate numerals from units, e.g., "30 µg" instead of "30µg."

6. Remove the redundant comma after "tested" in Line 116.

7. In Lines 123-125: "After quality control and DNA quantification by Nanodrop spectrophotometry and Qubit fluorometer [27], we denatured" — the phrase "we denatured" lacks an object, making it unclear what was denatured (e.g., DNA, RNA, protein, etc.). I suggest using either of the two statements: "After quality control and DNA quantification using Nanodrop spectrophotometry and Qubit fluorometer [27], the DNA was denatured." "After quality control and DNA quantification by Nanodrop spectrophotometry and Qubit fluorometer [27], we denatured the DNA."

8. Lines 192-195: The results in Table 1 demonstrate biochemical differences (e.g., CIT, TDA, and MEL positivity) between E. ruyisia FK53-34 and reference strains E. ruysiae AB136 and E. coli. However, the taxonomic significance of these differential biochemical characteristics was not extensively analyzed in the discussion. Could these distinct biochemical profiles potentially serve as diagnostic markers for E. ruysiae FK53-34?

9. The genome assembly demonstrates high-quality metrics, but the study lacks mention of PCR or Sanger sequencing validation for critical antimicrobial resistance/virulence genes. I recommend either: Supplementing with experimental validation data, or Citing established justification from similar studies (e.g., high coverage support rendering validation unnecessary.

10. The authors did not sufficiently explain in the Discussion why FK53-34 shows phenotypic susceptibility despite carrying resistance genes (e.g., blaEC-15, eptA, pmrF) in its genome. Did the authors examine the expression levels of these genes (e.g., by RT-qPCR)? I recommend discussing potential regulatory mechanisms (e.g., gene silencing or low expression) or citing similar studies to address this discrepancy.

11. The proposed association between virulence genes (e.g., aap/aspU, aatABCP, senB, traJT) and the fatal outcome in the pediatric case lacks direct evidence (e.g., histopathology or animal model data). The authors should clearly state this study limitation and suggest follow-up experimental approaches.

12. The authors should integrate direct accession links (e.g., BioProject PRJNA1223433) into the relevant Methods or Results sections, in addition to any supplementary material listings.

Comments on the Quality of English Language

The manuscript demonstrates high-quality writing with clear language expression and proficient English usage. However, as indicated above, certain sentences still require refinement.

Reviewer 2 Report

Comments and Suggestions for Authors

This study presents a whole-genome analysis of an Escherichia ruysiae strain isolated from a child in Togo presenting with gastroenteritis and bloody diarrhea.

  • Escherichia ruysiae was regarded as associating with gastroenteritis and bloody diarrhea in this study. Were any other potential pathogens identified in this case?
  • Please verify the text on lines 94-96, specifically the phrase "…in a 1:10 sample-to-broth ratio…".
  • Table 2 provides minimal additional information and could be omitted. The relevant data can be adequately described within the main text.
  • All figures currently have low resolution, which compromises their clarity and affects readability.
  • Please conduct a thorough review throughout the manuscript to ensure consistent and correct formatting of gene names (including antibiotic resistance genes) and bacterial taxonomic nomenclature.

Reviewer 3 Report

Comments and Suggestions for Authors

microorganisms-3815587

The authors describe the whole genome analysis of a single strain of E. ruysiae from a fatal case of gastroenteritis.

This is a highly problematic manuscript. At first glance, it looks fine, however, the more one goes into the details, things become more and more difficult. This is mainly due to the fact that the sequence of the strain deposited does not represent a closed genome, but consists of a total of at least 136 contigs of different sizes. The authors have done an online analysis of their sequences by publicly available bioinformatic tools, however without questioning the results obtained. Another problem comes from the low coverage of 20%, which tells that the sequence may not be particularly reliable.

  1. For the analysis of antimicrobial resistance genes, they used five different databases (Resfinder, NCBI, CARD, ARG-ANNOT, and MEGAres). This analysis identified several beta-lactamase genes, including blaPBP, blaAmpH, and blaEC-15. Unfortunately, the authors did not say anything about the percentage of identity of their bla genes to the reference bla genes in the respective databases. Here it is important that the genes in question should have the same length as the reference genes and the identity should be higher than 95%. I have redone this analysis, and only CARD identified solely blaEC-15 … however, with a low coverage of less than 90%.
  2. I am not sure whether blaPBP is at all a resistance gene. As this gene is present in all E. ruysiae – regardless of phenotypic beta-lactam resistance –, I assume, it is the gene for a penicillin-binding protein. Here it is necessary to know that mutations in genes coding for penicillin-binding proteins have been described to result in beta-lactam resistance, but PBP-encoding genes are not always involved in beta-lactam resistance genes. The AmpH genes are also present in all E. ruysiae strains, again regardless of beta-lactam resistance. Maybe this gene is involved in the intrinsic resistance of Escherichia to penicillin.
  3. Despite the fact that there seem to be several bla genes present in strain FK53-34, none of them seems to be functional in conferring resistance to any of the tested beta-lactams (not even amoxicillin). Are all the genes complete. Do they have intact promoters, ribosome binding sites etc.?
  4. There are several other genes mentioned as being “resistance genes”, e.g. acrA, emrF, baeS, and tolC. These genes are not real resistance genes, but code for components of mainly RND-type multidrug transporters that – under certain circumstances (e.g. when overexpressed) – can also export the one or the other antimicrobial agent. The authors need to avoid to consider them as real resistance genes.
  5. Finally, the genes eptA (synonym: pmrC) and pmrF modify lipid A and thereby cause a decreased binding of polymyxins (polymyxin B, colistin).
  6. I wonder why the authors have only tested their strain for phenotypic susceptibility to beta-lactams and (fluoro)quinolones. The testing of colistin and /or polymyxin B would have given at least a hint towards the activity of the genes eptA and pmrF.
  7. In the antimicrobial susceptibility testing (AST) part, there is no information about the quality control strain included in AST. The testing of a suitable QC strain is indispensable. Without QC data, the AST results cannot be considered as valid.
  8. Lines 112-116: Cefepime, Cefuroxime, Ceftazidime and Cefotaxime are misspelled. According to EUCAST, a 5 µg cefotaxime disk should be used, not a 30 µg disk. For Nalidixic acid, amoxicillin and ticarcillin, EUCAST does not provide suitable disk contents in µg. The method is called agar disk diffusion – not Kirby-Bauer method.
  9. There is a statement about the mobilome of strain FK53-34 in lines 255-257. This statement “The mobilome is composed of 106 integrations/excisions, 167 phages, 86 Transposons, 126 replication:recombinations, RRR repair, and 70 stability, transfer and defense MST.” is not supported by any data. It is hard to believe that 167 phages have been detected in this one genome – bearing in mind that phages are usually at least 40 kb in size, this would end up in a size of at least 6.680 Mb only for the phages although the whole genome is less than 5 Mb. So, there must be something wrong. Moreover, the authors claim to have identified 86 transposons. This is also a rather high number – does this include real transposons and also insertion sequences? The authors need to specify the types of transposons and the types of insertion sequences. How have the numbers of 106 integrations/excisions be determined and in comparison to which reference genome. The same is true for 126 replication:recombinations. In this connection what does the abbreviation MST stand for – it has not been explained.
  10. Table 2 is unnecessary – the information provided in this table has already been mentioned in the text (lines 196-198)
  11. Figures 1 and 2 are too small and too fuzzy – they are hardly readable  – even in the highest magnification. Figures with a distinctly higher resolution are necessary.
  12. Table 3: Does this table refer only to E. ruysiae strains for which whole genome sequences are available? Here tet(A) should be written with (A) in round brackets and not in italics. Please not that tetR is not a resistance gene, but a tet(A)-associated repressor gene. CTX should read blaCTX-M with CTX-M as subscript. Moreover, the number of the blaCTX-M-variant needs to be indicated. The gene qnrS1 is written without a hyphen.
  13. Table 4 should be deleted. The content of this table can be included in 1-2 sentences in the text.
Comments on the Quality of English Language

There are here and there some strange sentences, spelling mistakes, mistakes in punctation etc.